# DyST: Towards Dynamic Neural Scene Representations on Real-World Videos

**Maximilian Seitzer** [1]* **Sjoerd van Steenkiste** [2] **Thomas Kipf** [3]

**Klaus Greff** [3] **Mehdi S. M. Sajjadi** [3]

[1]MPI for Intelligent Systems [2]Google Research [3]Google DeepMind

## Abstract

Visual understanding of the world goes beyond the semantics and flat structure of individual images. In this work, we aim to capture both the 3D structure and dynamics of real-world scenes from monocular real-world videos. Our Dynamic Scene Transformer (DyST) model leverages recent work in neural scene representation to learn a latent decomposition of monocular real-world videos into scene content, per-view scene dynamics, and camera pose. This separation is achieved through a novel co-training scheme on monocular videos and our new synthetic dataset DySO. DyST learns tangible latent representations for dynamic scenes that enable view generation with separate control over the camera and the content of the scene.

## 1 Introduction

The majority of research in visual representation learning focuses on capturing the semantics and 2D-structure of individual images. In this work, we instead focus on simultaneously capturing the *3D structure* as well as the *dynamics* of a scene, which is critical for planning, spatial and physical reasoning, and effective interactions with the real world. We draw upon the recent progress in generative modelling of 3D visual scenes, which has moved away from explicit representations such as voxel grids, point-clouds or textured meshes in favor of learning implicit representations by directly optimizing for novel view synthesis (NVS). For example, Neural Radiance Fields, though initially limited to single scenes with hundreds of input images with controlled lighting, precise camera pose and long processing times (Mildenhall et al., 2020), have since been extended to handle variations in lighting (Martin-Brualla et al., 2021), generalize across scenes (Trevithick & Yang, 2021), work with few images (Niemeyer et al., 2022), missing cameras (Meng, 2021), and even dynamic scenes (Pumarola et al., 2020).

A related line of research focuses on learning global *latent* neural scene representations (Sitzmann et al., 2021; Sajjadi et al., 2022b; Kosiorek et al., 2021). These approaches offer several advantages, including the ability to generalize from few views, and improved scalability and efficiency due to amortized learning (Sajjadi et al., 2022b). Most importantly, their tangible latent representations can be readily used for downstream applications (Sajjadi et al., 2022a; Driess et al., 2022; Jabri et al., 2023). However, each of these models is limited to *static* scenes, which not only ignores the important aspect of scene dynamics, but also disqualifies the vast majority of potential real-world video datasets that contain dynamic scenes.

In this work, we present first steps towards learning *latent dynamic neural scene representations* from monocular real-world videos. Building up on recent work, our method learns a separation of the scene into global content and per-view camera pose & scene dynamics, thereby enabling independent control over these factors.

Our core contributions are as follows:

- We propose the *Dynamic Scene Transformer (DyST)*, a model that learns latent neural scene representations from monocular video and provides controlled view generation.

---

*Work done during an internship at Google DeepMind. Project website: dyst-paper.github.io.

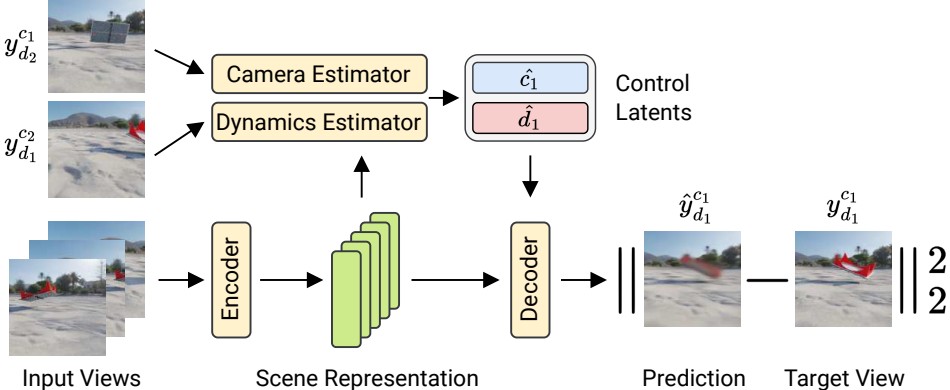

Figure 1: The DyST model. Input views of a scene are encoded into the scene representation $\mathcal{Z}$ capturing the scene content. The model is trained with an $L_2$ loss by synthesizing novel target views from $\mathcal{Z}$. To identify the target view $y_{d_1}^{c_1}$, the camera and dynamics estimators produce the low-dimensional camera control latents $\hat{c}_1, \hat{d}_1$ from *views with matching camera ($y_{d_2}^{c_1}$) and dynamics ($y_{d_1}^{c_2}$)*. This scheme, termed *latent control swap*, induces a separation of camera and scene dynamics in the latent space (see Sec. 3.1). We co-train DyST on synthetic multi-view scenes and real-world monocular video, transferring the latent structure and thereby enabling controlled generation on real videos.

- Through a unique training scheme, DyST learns a latent decomposition of the space into *scene content* as well as per-view *scene dynamics* and *camera pose*.
- We present a detailed analysis of the model and its learned latent representations for scene dynamics and camera pose.
- Finally, we propose DySO, a novel synthetic dataset used for co-training DyST. We will publish DySO to the community for use in co-training and evaluation of future work on dynamic neural scene representations.

## 2 RELATED WORK

The field of *neural rendering* for static 3D scenes has recently experienced a significant gain in popularity with the introduction of Neural Radiance Field (NeRF) (Mildenhall et al., 2020) which optimizes an MLP through the novel view synthesis (NVS) task on posed RGB imagery. NeRF and similar scene-specific methods typically require a very dense coverage of the scene along with accurate camera pose information. While there exists a great deal of works tackling these shortcomings (e.g. Yu et al., 2021; Meng, 2021; Niemeyer et al., 2022), this line of research largely focuses on view synthesis quality rather than learning latent scene representations. We refer the reader to Tewari et al. (2022) for a recent overview.

More closely related to our approach are methods that learn 3D-aware *latent* scene representations from the NVS task. Examples in this line of work include GQN (Eslami et al., 2018), NeRF-VAE (Kosiorek et al., 2021), LFN (Sitzmann et al., 2021), and SRT (Sajjadi et al., 2022b). These methods combine a visual encoder with a camera pose-conditioned decoder network for novel view synthesis. More recently, Sajjadi et al. (2023) proposed RUST, which uses a Pose Estimator to train the model without any camera pose information. Contrary to DyST, all of these methods are limited to *static* scenes, and, with the exception of RUST, require accurate camera poses for training.

Learning scene representations from *dynamic* scenes comes with additional challenges: in the absence of calibrated multi-camera setups that record posed multi-view video, models have to learn to capture and disentangle scene dynamics from camera motion. In the easier case of posed multi-view video, high-fidelity solutions exist, such as D-NeRF (Pumarola et al., 2020), H-NeRF (Xu et al., 2021), NeRF-Dy (Li et al., 2022), and NeRFlow (Du et al., 2021). Acquiring the right data for these methods requires specialized recording setups and is not applicable to abundantly available monocular video. To overcome this limitation, several works including Neural Scene Flow Fields (Li et al., 2021), Spacetime Neural Irradiance Fields (Xian et al., 2021), TöRF (Attal et al., 2021), and Gao et al. (2021)

make use of additional priors obtained from optical flow or depth to support NVS for small camera displacements directly from monocular video. NerFPlayer (Song et al., 2023) and RoDynRF (Liu et al., 2023) are able to model dynamic scenes without depth or optical flow information by learning separate NeRFs for dynamic and static parts of the scene, though they require training a separate model for every video without generalizing between scenes. MonoNeRF (Fu et al., 2023) learns generalizable scene representations from pose-free monocular videos, but different from our method, it assumes a static underlying scene, i.e. it is not applicable to videos of dynamic scenes.

## 3 METHOD

**Dynamic Scenes.** A *dynamic scene* $\mathcal{X}$ consists of an arbitrary number of images or views $x$ with associated *camera* pose $c_i$ and *scene dynamics* $d_j$, i.e. $\mathcal{X} = \{x_{d_1}^{c_1}, x_{d_2}^{c_2}, \ldots\}$. The cameras $c_i$ define the extrinsics and intrinsics of the camera that this particular image has been taken with, while the dynamics $d_j$ define the position and shape of entities in the scene, e.g. the position of a moving car.

We note two special cases thereof: in *static scenes*, there exist no scene dynamics, hence only the camera view varies across views: $\mathcal{X} = \{x_d^{c_1}, x_d^{c_2}, \ldots\}$. This static-scene assumption significantly simplifies the setting by allowing the usage of basic photogrammetric constraints (Mildenhall et al., 2020), and it forms the basis for the majority of prior works on learned implicit representations (Sitzmann et al., 2021; Sajjadi et al., 2022b; Kosiorek et al., 2021).

Common real-life *videos* are monocular, i.e. both scene dynamics and camera vary together, but are strongly correlated over time, i.e. $\mathcal{X} = \{x_{d_t}^{c_t}, \ldots\}$ for time step $t = 1, \ldots, T$, complicating the learning of disentangled representations from this source of data. Our goal is to learn *latent neural scene representations* on real-life video data while gaining *independent* control over both scene dynamics and the camera position.

**Neural Scene Representations.** Given a dynamic scene $\mathcal{X}$, we select a number of *input views* $X = \{x, \ldots\} \subset \mathcal{X}$ that are encoded by the model into a set-based neural scene representation

$$\mathcal{Z} = \{\boldsymbol{z}_k \in \mathbb{R}^d\}_k = \text{Enc}_\theta(X). \tag{1}$$

We note that this setting is in contrast to most common approaches based on Neural Radiance Fields (NeRF) (Mildenhall et al., 2020), which require training a model per scene and do not provide *latent* neural scene representations beyond the scene-specific MLP weights. Following Sajjadi et al. (2022b), the encoder $\text{Enc}_\theta$ first applies a convolutional neural network to each input view $x \in X$ independently, then flattens the resulting feature maps into a set of tokens that is jointly processed with an encoder transformer (Vaswani et al., 2017).

To reproduce a novel *target view* $y_d^c \in \mathcal{X}$ of the same scene, we use a transformer decoder $\text{Dec}_\theta$ that repeatedly cross-attends into the scene representation $\mathcal{Z}$ to retrieve information about the scene relevant for this novel view. In order to do so, the decoder needs to be informed of the desired target view *camera pose* $c$ and *dynamics* $d$. Most existing work only covers *static* scenes (Sitzmann et al., 2021; Kosiorek et al., 2021; Sajjadi et al., 2022b), and simply assumes a known ground-truth camera position $c$ for querying novel views:

$$\hat{y}^c = \text{Dec}_\theta(c, \mathcal{Z}). \tag{2}$$

**Inferred latent camera poses.** Access to the ground-truth camera poses is a strong assumption that does regularly not extend to real-world settings. In practice, camera parameters are often estimated using external sensors such as LIDAR, or from the RGB views through Structure-from-Motion (SfM) methods such as COLMAP (Schönberger & Frahm, 2016). However, these methods are generally noisy and regularly fail altogether (Meng, 2021; Sajjadi et al., 2023), especially so in dynamic scenes such as natural videos.

To lift any requirements for explicit camera poses on static scenes, Sajjadi et al. (2023) introduce a *Camera Estimator* module $\text{CE}_\theta$ that learns to extract a *latent camera pose* $\hat{c}$ from the RGB data itself. More specifically, the camera estimator receives the target view $y$ as input and queries parts of the scene representation $\mathcal{Z}' \subset \mathcal{Z}$ to produce a low-dimensional camera control latent

$$\hat{c} = \text{CE}_\theta(y, \mathcal{Z}') \in \mathbb{R}^{N_c}. \tag{3}$$

In practice, $\mathcal{Z}'$ contains tokens belonging to the first input view, encouraging the learned camera control latent to be relative to that specific view. We note that $\hat{c}$ lives in an abstract feature space, and it is not immediately compatible with explicit GT camera poses $c$, though such a mapping can be learned from data (Sajjadi et al., 2023). This estimated camera control latent is then used to condition the decoder when generating an image during training, i.e. $\hat{y}^c = \mathrm{Dec}_\theta(\hat{c}, \mathcal{Z})$, entirely removing the need for camera poses to train such a model. Sajjadi et al. (2023) show that the learned camera control latent distribution accurately follows the true poses, and that this model, on static scenes, achieves similar generation quality as baselines trained with ground truth pose information.

The network architecture of the camera estimator is similar to the encoder's: a CNN processes the target view, followed by a transformer that alternates between cross-attending into the scene representation, and self-attending on the flattened feature maps. The final camera control latent $\hat{c}$ is then produced by a global average pooling of the output tokens and a linear projection to $N_c$ dimensions.

### 3.1 Structuring the latent space for dynamic scene control

Up until this point, the setting in Eqs. (2) and (3) only admits static scenes. To apply our method on real-world videos, we need a way to allow variations in scene dynamics. We start with an approach similar to the learned camera control latents: a *Dynamics Estimator* DE sees the target view $y$ and $\mathcal{Z}'$, and learns to extract a *dynamics control latent* $\hat{d}$ from the RGB views:

$$\hat{d} = \mathrm{DE}_\theta(y, \mathcal{Z}') \in \mathbb{R}^{N_d}. \tag{4}$$

The dynamics control latent $\hat{d}$ is used as an additional query for the decoder to reconstruct the target view: $\hat{y}^c_d = \mathrm{Dec}_\theta(\hat{c}, \hat{d}, \mathcal{Z})$.

**Separating dynamics from camera pose.** While this split between camera pose and scene dynamics is convenient in theory, we note that thus far, there exist no structural differences between the ways in which $\hat{c}$ and $\hat{d}$ are inferred or used. Hence, in practice, they likely will both learn to capture the camera pose and scene dynamics in a unified way, making it hard to control these aspects independently (as we confirm in practice in Sec. 4.4).

We propose to enforce this separation through a novel *latent control swap* training scheme that is outlined in Fig. 2. For training, we assume a scene $\mathcal{X}$ where views are available for all combinations of camera poses $\{c_i\}_i$ and scene dynamics $\{d_i\}_i$. As target frames, we choose the four views corresponding to all combinations of the camera poses $c_1, c_2$ and dynamics $d_1, d_2$, i.e. our set of target views is $\{y^{c_1}_{d_1}, y^{c_2}_{d_1}, y^{c_1}_{d_2}, y^{c_2}_{d_2}\}$. For all remaining camera poses and scene dynamics, we randomly sample a subset of views as input views.

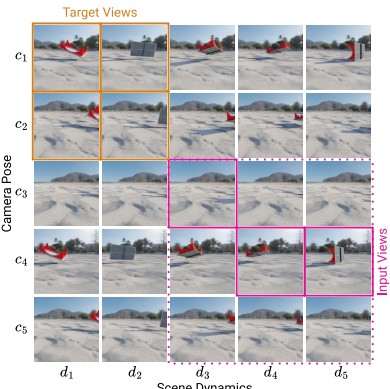

Figure 2: Illustration of a DySO scene.

In the following, we describe the decoding mechanism for the target view $y^{c_1}_{d_1}$; the scheme for the remaining target views follows analogously. To generate this particular target view, the decoder needs access to (a latent estimate for) the camera pose $c_1$ and the scene dynamics $d_1$. Our key insight is that we do not necessarily need to estimate $\hat{c}_1$ and $\hat{d}_1$ from the target frame $y^{c_1}_{d_1}$ itself. Instead, we can estimate the camera pose and scene dynamics from their respective counterparts:

$$\hat{c}_1 = \mathrm{CE}_\theta(y^{c_1}_{d_2}, \mathcal{Z}'), \quad \hat{d}_1 = \mathrm{DE}_\theta(y^{c_2}_{d_1}, \mathcal{Z}'). \tag{5}$$

Crucially, we note that the most salient information $y^{c_1}_{d_2}$ contains to render the target view $y^{c_1}_{d_1}$ is the camera pose and it is hence natural for the camera estimator CE to learn to extract camera pose information from its inputs. Similarly, $y^{c_2}_{d_1}$ shares its scene dynamics $d_1$ with the target view to be generated, encouraging the dynamics estimator DE to extract scene dynamics. From these estimates, we render the target view:

$$\hat{y}^{c_1}_{d_1} = \mathrm{Dec}_\theta\left(\mathrm{CE}_\theta(y^{c_1}_{d_2}, \mathcal{Z}'), \mathrm{DE}_\theta(y^{c_2}_{d_1}, \mathcal{Z}'), \mathcal{Z}\right). \tag{6}$$

As a consequence, the model is required to route all information regarding the camera pose through $\hat{c}_1$, and all information regarding the scene dynamics through $\hat{d}_1$, yielding the desired separation.

## 3.2 SIM-TO-REAL TRANSFER

The approach outlined requires the availability of a special kind of multi-view, multi-dynamics dataset. In the real world, multi-view video would satisfy the requirements: the scene dynamics are synced through time, while each separate video instantiates a different view onto the same scene. However, acquiring such a dataset for training would narrow the applicability of the method to a limited set of domains. Instead, we suggest a sim-to-real setup: we generate a synthetic dataset that contains the desired properties for inducing camera-dynamics separation in the learned latent representations, while co-training on natural, monocular videos.

**Synthetic dataset.** We follow prior work using Kubric (Greff et al., 2022) and ShapeNet objects (Chang et al., 2015). Our dataset uses components from the MultiShapeNet (MSN) dataset (Sajjadi et al., 2022b) that has been introduced as a challenging test bed for 3D representation learning from static scenes. MSN uses photo-realistic ray tracing, high resolution images as backgrounds, and has a diverse set of realistic 3D objects.

Compared to MSN, our dataset has some distinct modifications. We place one object into each scene, as our primary goal is the separation of scene dynamics, in this case the object position & orientation, from the camera. The object is initialized with a random position and pose on the floor. To integrate dynamics, for each time step, we randomly jitter the object's position and further apply a random rotation to the object. MSN samples cameras on a half-sphere around the scene. Since real videos rarely feature 360 degree views, we use a set of more realistic camera motions: horizontal shifts, panning, and zooming, as well as sampling random camera points nearby a fixed point.

We find this simple set of camera and object motions to be sufficient for successful sim-to-real transfer, and did not find the need to simulate more physically realistic object motions. Our dataset consists of 1M dynamic scenes, each generated with all 25 combinations of 5 distinct scene dynamics and 5 camera views. We call this new dataset *Dynamic Shapenet Objects* (DySO) and will make it publicly available, as we expect it to be a useful tool for the community for training and evaluating dynamic neural scene representations in the future.

**Co-training.** To transfer the separation of dynamics and camera control latents from the synthetic scenes to natural monocular videos, we *co-train* on both types of data at the same time. More specifically, we take alternating optimization steps on batches sampled from each of the two datasets. On the synthetic batches, we train the model according to Eq. (6), i.e. the camera pose and scene dynamics are estimated from views $y_{d_2}^{c_1}$ and $y_{d_1}^{c_2}$, different from the actual target view $y_{d_1}^{c_1}$ to be rendered. On the batches with monocular videos, those additional views are not available, so we simply use the target view itself for dynamics and camera pose estimation:

$$\hat{y}_d^c = \text{Dec}_\theta\left(\text{CE}_\theta(y_d^c, \mathcal{Z}'), \text{DE}_\theta(y_d^c, \mathcal{Z}'), \mathcal{Z}\right). \tag{7}$$

The latent control swap outlined in Sec. 3.1 is implemented efficiently by estimating the cameras and dynamics for all four (randomly sampled) target views in parallel before swapping the estimated $\hat{c}_i$ and $\hat{d}_j$ control latents accordingly and decoding all four target views jointly.

**Model architecture.** Our network architecture largely follows Sajjadi et al. (2023). For efficiency, we implement the Camera and Dynamics Estimators CE & DE as a single transformer that produces $\hat{c}$ and $\hat{d}$ simultaneously for a given target view (see also Fig. 8). The two control latents are produced differently: for the camera pose, we apply global average pooling to the transformer's output, and linearly project the result to produce $\hat{c}$. For the scene dynamics, we add a learned token to the transformer's set of inputs, and apply a separate linear projection to $\hat{d}$ to only the output for that token. This follows the intuition that the camera pose is a global property affecting all tokens (hence global pooling), whereas estimating the dynamics is a more localized task that only affects object tokens (hence the additional token that can learn to attend to the respective subset of tokens). For the decoder Dec, we simply concatenate the two control latents to form a single *query* $[\hat{c}, \hat{d}]$ which is used to cross-attend into the SLSR $\mathcal{Z}$ to generate the target view.

All model weights $\theta$ are optimized end-to-end using the novel view synthesis (NVS) objective. Given a dataset $\mathcal{D} = \{\mathcal{X}_i\}_i$ of training scenes, for an arbitrary ground truth target view $y_d^c \in \mathcal{X}_i$, the model is trained to minimize the $L_2$ loss:

$$\mathcal{L}_{\text{NVS}}(\theta) = \mathbb{E}_{\mathcal{X} \sim \mathcal{D}, (X, y_d^c) \sim \mathcal{X}} \left[\|\text{Dec}_\theta(\text{CE}_\theta(y_{d'}^c, \mathcal{Z}'), \text{DE}_\theta(y_d^{c'}, \mathcal{Z}'), \text{Enc}_\theta(X)) - y_d^c\|_2^2\right], \tag{8}$$

where $c' \neq c$, $d' \neq d$ if $\mathcal{X}$ is synthetic, and $c' = c$, $d' = d$ otherwise, i.e. on real-world videos.

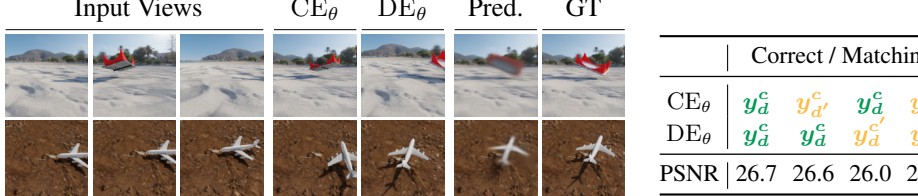

| | Correct / Matching | | | | Wrong | |
|---|---|---|---|---|---|---|
| $\mathrm{CE}_\theta$ | $y_d^c$ | $y_{d'}^c$ | $y_d^c$ | $y_{d'}^c$ | $y_d^{c'}$ | $y_d^c$ |
| $\mathrm{DE}_\theta$ | $y_d^c$ | $y_d^c$ | $y_d^{c'}$ | $y_d^{c'}$ | $y_d^c$ | $y_{d'}^c$ |
| PSNR | 26.7 | 26.6 | 26.0 | 26.0 | 18.3 | 22.3 |

Figure 3: NVS on DySO. **Left:** Qualitative results. DyST is able to learn how to extract camera & dynamics independently from the respective views of the scene, leading to the correct prediction of the GT image. **Right:** Quantitative performance for various inputs for $\mathrm{CE}_\theta$ & $\mathrm{DE}_\theta$. PSNR is high even when camera or dynamics are only from matching views, showing that the model is capable of estimating camera & dynamics independently of the other.

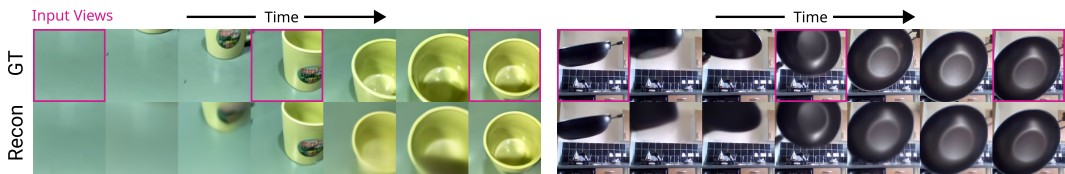

Figure 4: Frame synthesis on SSv2. We use the first, middle, and last frame as input views (marked in purple), and generate the intermediate frames based on the control latents estimated from them. DyST is able to render videos with challenging camera (left) and object motions (right).

## 4 EXPERIMENTS

Unless specified otherwise, DyST is always co-trained on our synthetic DySO dataset and on real world videos as described in Secs. 3.1 and 3.2. As a source of a real world videos, we use the *Something-Something v2* dataset (SSv2) (Goyal et al., 2017), which consists of roughly 170K training videos of humans performing basic actions with everyday objects. The videos contain both nontrivial camera motion and dynamic scene manipulation. From each video, we sub-sample a clip of 64 frames. The model is trained using 3 randomly sampled input frames $X$ to compute the scene representation $\mathcal{Z} = \mathrm{Enc}(X)$ and 4 randomly sampled target views for reconstruction. Following Sajjadi et al. (2022b), we train using a batch size of 256 for 4M steps. For both control latents, we use a dimensionality of $N_c = N_d = 8$. We refer to Appendix A.1 for further implementation details.

### 4.1 NOVEL VIEW SYNTHESIS

We begin our investigations by testing the novel view synthesis (NVS) capabilities of the model on the test splits of both the synthetic DySO and the real SSv2 datasets. We show several examples in Fig. 3 (left). On DySO, the background is particularly well-modelled, while the object is often less sharp. We attribute this to the difficulty of capturing 3D object appearance from only 3 provided input views. For SSv2, we use the first, middle, and last frame of the videos as input frames and generate intermediate frames using control latents estimated from the intermediates. Qualitative results are shown in Fig. 4. We find that our model accurately captures changes in both camera viewpoint and object dynamics.

Quantitatively, we evaluate the model on DySO by computing the PSNR against the ground truth target view $y_d^c$ while varying the control latents, see Fig. 3 (right). We observe that there is only a minor difference in PSNR between using the target $y_d^c$ itself to estimate the control latents ($y_d^c/y_d^c$), versus estimating the latent camera from a view with different scene dynamics ($y_{d'}^c$) and vice versa ($y_d^{c'}$). This further confirms that DyST can successfully estimate and re-combine camera pose and scene dynamics to synthesize the desired novel views.

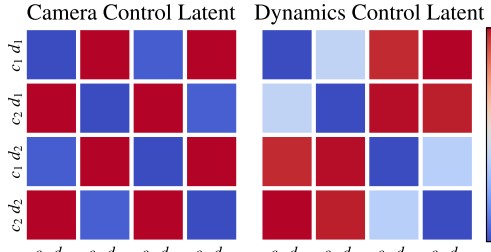 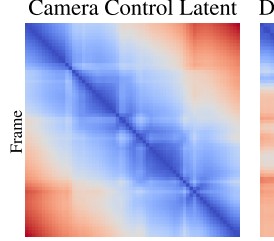 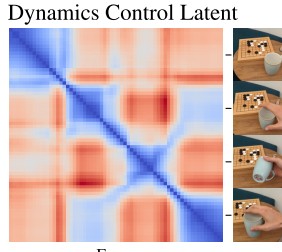

Figure 5: Latent distance analysis. **Left:** average L2 distances in the latent space between pairs of views on DySO, for camera (left) and dynamics control latents (right). **Right:** frame-to-frame L2 distances for a real world video, for camera (left) and dynamics control latents (right). The distances closely follow events in the video: grasping (second frame), turning (third frame) and placing the cup (last frame) are visible as distinct areas in the dynamics latent distances. The slow panning camera movements is reflected in the broad diagonal stripe for the camera latent distances. Notably, the grasp has low distance to the similar placement motion despite the different camera positions, indicating that the model has learned to encode dynamics independently of the camera pose.

## 4.2 LEARNED CAMERA AND SCENE DYNAMICS CONTROL LATENTS

In this section, our experiments aim to answer three questions. First, does our training with latent control swapping induce a meaningful separation of camera pose and scene dynamics on synthetic data? Secondly, does the latent space separation transfer to real videos? And finally, what is the structure of the learned latent spaces?

**Separation of Camera and Dynamics.** To analyze the separation into camera and scene dynamics quantitatively, we develop a metric which we call *contrastiveness*. Let $D^{\mathrm{cam}}(y, y', \mathcal{Z}') = \|\mathrm{CE}(y, \mathcal{Z}') - \mathrm{CE}(y', \mathcal{Z}')\|$ denote the L2 distance in camera latent space between views $y$ and $y'$. $D^{\mathrm{dyn}}$ is defined accordingly for the dynamics latent space. Then, we compute the average ratio $R^{\mathrm{cam}}$ between the distance in camera latent space of a reference view $y_d^c$ to a matching view $y_{d'}^c$, and to a non-matching view $y_d^{c'}$:

$$R^{\mathrm{cam}} = \mathbb{E}_{\mathcal{X} \sim \mathcal{D},\, (X, y_d^c, y_{d'}^c, y_d^{c'}) \sim \mathcal{X}} \left[ \frac{D^{\mathrm{cam}}(y_d^c, y_{d'}^c, \mathcal{Z}')}{D^{\mathrm{cam}}(y_d^c, y_d^{c'}, \mathcal{Z}')} \right], \quad \text{with } c \neq c' \text{and } d \neq d'. \quad (9)$$

Analogously, we define $R^{\mathrm{dyn}}$ using $y_d^{c'}$ as matching and $y_{d'}^c$ as non-matching view. If the contrastiveness metric $R^{\mathrm{cam}}$ is $\approx 0$, this indicates that views with the same camera are mapped onto the same point in camera latent space regardless of their dynamics; if $R^{\mathrm{cam}}$ is $\approx 1$, views with non-matching camera can not be distinguished based on latent distance on average. If $R^{\mathrm{cam}}$ is $> 1$, information from the dynamics latent space is "leaking" into the camera latent space, because views with the same dynamics $d$ but different camera $c'$ are on average closer to $y_d^c$ than views with different dynamics $d'$, but the same camera $c$.

We find that the learned latent spaces are well separated on average, with little influence of one control latent on the other. In particular, our model achieves a camera contrastiveness of $R^{\mathrm{cam}}$=0.06 (corr. to $16.7\times$ less distance of matching to non-matching views), and a dynamics contrastiveness of $R^{\mathrm{dyn}}$=0.42 (corr. to $2.4\times$ less distance). The fact that $R^{\mathrm{cam}}$ is notably smaller than $R^{\mathrm{dyn}}$ indicates that it is easier for the model to estimate the camera pose compared to the object pose. This might be intuitive, as the camera pose can be estimated directly from all background pixels (unaffected by the scene dynamics), while the appearance of scene dynamics in 2D images is the result of a combination of the actual scene dynamics *and* the camera pose. In contrast, when training without latent control swapping, the model only achieves $R^{\mathrm{cam}}$=0.72 and $R^{\mathrm{dyn}}$=1.26, indicating that the separation is drastically worse there (see also Tab. 1). This is further evidence that our latent control swap training scheme induces the desired disentanglement of camera pose and scene dynamics. We also visualize the average latent distances as a heatmap in Fig. 5, and find that matching (blue squares) and non-matching views (red squares) are easily distinguishable.

**Transfer to Real Video.** To see whether the camera-dynamics separation transfers to real videos, we measure the frame-to-frame latent distances on an example video of one of the authors grasping,

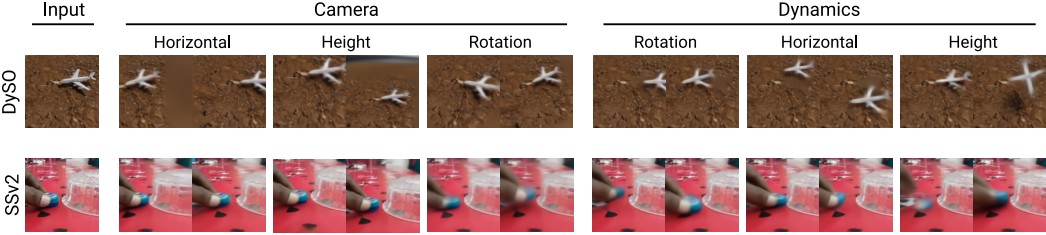

Figure 6: Interpolating along the principal components of camera and scene dynamics latent spaces on the synthetic DySO (first row) and the real world SSv2 dataset (second row) reveals that the model learns meaningful, isolated motions for both camera and dynamics.

lifting and turning a coffee cup, before placing it back in its original position. The resulting distance matrix is visualized in Fig. 5. We find both the slow panning motion of the camera and the distinct events in the video are clearly visible in the distance maps. Moreover, we point out that similar object poses lead to similar dynamics latents despite the vastly different camera positions. We conclude that the model is able to successfully disentangle camera pose from scene dynamics in real-world videos.

**Learned Structure.** To investigate the structure of the two learned control latent spaces, we apply principal component analysis (PCA) to a set of latent cameras $\hat{c}$ and scene dynamics $\hat{d}$ collected from 300 test scenes of the DySO dataset. Starting from the extracted latents $\hat{c}$ & $\hat{d}$ of some view, we then linearly interpolate along the principal components of each space and show the decoded images in Fig. 6. We find that the learned spaces capture meaningful movements: the first three components for the camera pose capture horizontal and vertical movements and planar rotation, mirroring similar observations by Sajjadi et al. (2023). For the scene dynamics, the first component captures object rotation, the second represents horizontal and the third vertical movements. Furthermore, we observe additional evidence that camera and dynamics are well separated: changing $\hat{c}$ does not modify the object's pose; similarly, changing $\hat{d}$ does not modify the camera. Lastly, we also estimate the principal components on the SSv2 dataset, and find that they capture similar effects for the camera; for the dynamics, movements are more constrained to plausible dynamics fitting to the input frames.

We conclude that our model is able to disentangle camera and scene dynamics, and that it learns a meaningful structure for both. We study how these capabilities can be used for test-time video control in the next section.

## 4.3 CONTROLLABLE VIDEO MANIPULATION

After establishing that DyST learns a disentangled representation of camera and scene dynamics, we now show how this capability can be used to manipulate real-world video post-hoc. After encoding a target video into the scene representation $\mathcal{Z}$ using the the first, middle and last frame, we synthesize counterfactual variations for this video by manipulating the control latents $\hat{c}$ & $\hat{d}$. We study two variants: motion freezing and video-to-video motion transfer. For the former, we compute $\hat{c}$ or $\hat{d}$ from a frame of the target video, and keep it fixed throughout the video. This freezes the corresponding camera or object motion from that frame throughout the video, creating a "bullet time" effect. For the latter, we replace $\hat{c}$, $\hat{d}$, or both, with the latents computed from a different video, effectively transferring the motion in that video to the target frame or video.

Figure 7 shows examples of such manipulations. First, we find that inserting the latent camera $\hat{c}$ or scene dynamics $\hat{d}$ of a source frame correctly keeps the camera / object stable throughout the video, but with the original object / camera trajectory. Second, we find that transferring the camera from a video with a camera zoom or horizontal pan copies the corresponding motion onto a target frame. Note how on the shift motion, the left edge of the frame becomes blurred — the respective areas are not visible in the target video, and thus DyST fills in an average guess due to the L2 loss. It is notable that both types of manipulations work in this fashion, as the manipulated control latents are fully out-of-distribution for the decoder given the scene representation.

Figure 7: Video manipulations. **Left**: we re-generate the video (row 1) while fixing the camera $\hat{c}$ (row 2) or dynamics control latent $\hat{d}$ (row 3) to one estimated from two frames of the video (columns 1 & 2) throughout. The fixed camera or dynamics appear frozen in place, whereas the corresponding other motion is unchanged. **Right**: transferring the camera pose $\hat{c}$ of source videos with a zoom (row 1) and a horizontal pan (row 2) to a target frame, creating a video with the respective camera motions.

| Method | ↑PSNR | ↓LPIPS | ↓$R^{\text{cam}}$ | ↓$R^{\text{dyn}}$ |
|---|---|---|---|---|
| No swap | 18.6 | 0.45 | 0.72 | 1.26 |
| 50% swap | 25.4 | 0.36 | 0.26 | 1.07 |
| Latent averaging | 22.9 | 0.42 | 0.54 | 0.96 |
| DyST | **26.0** | **0.34** | **0.06** | **0.42** |

Table 1: Evaluation of alternative training techniques evaluated for the $y_{d'}^{c}/y_d^{c'}$ case (Fig. 3) on DySO. The alternatives perform worse across all image-based and contrastiveness-based metrics, showing that our *latent control swap* approach (Sec. 3.1) leads to the best separation between latent camera and dynamics.

## 4.4 ABLATION STUDY

We investigate the importance of the latent control swap (Eq. (6)). We measure PSNR under latent control swapping as described in Sec. 4.1, and latent space contrastiveness as described in Sec. 4.2 on the DySO dataset. The results are listed in Tab. 1. We compare several variations: applying no swap at all (i.e., decoding the target $y_{d_1}^{c_1}$ using control latents of $y_{d_1}^{c_1}$ itself), swapping randomly with a 50% probability, or averaging the control latents of the target view and the swapped view together (e.g., averaging camera latents from $y_{d_1}^{c_1}$ and $y_{d_2}^{c_1}$ to decode $y_{d_1}^{c_1}$). The latter is motivated by the idea that the latents of matching views should eventually converge to the same point. We find that all variations perform significantly worse than the latent control swap: without swapping, latent space separation does not occur as expected because there is no structural incentive that enforces it — consequently, the recombination PSNR is low. When swapping only 50% of the time, some separation emerges, but the model still has some chance to mix information from the target view into both control latents. For averaging the latents, we also find substantial entanglement of camera and dynamics. Thus, our latent control swap is the most suitable training scheme for consistent separation of camera and dynamics in the latent space.

## 5 CONCLUSION

In this work, we propose DyST, a novel approach to generative modeling of dynamic 3D visual scenes that admits separate control over the camera and the content of the scene. We demonstrate how DyST can be applied to real-world videos of dynamic scenes despite not having access to ground-truth camera poses via sim-to-real transfer. DyST shows promising view synthesis and scene control capabilities on real-world videos of dynamic scenes.

As a next step, we would like to apply our method to more complex types of videos, for example with several independent moving objects, longer camera trajectories, or changing lighting conditions — challenges which exceed the scope of this work. Indeed, we expect that further model innovations are needed to tackle them. Another open problem concerns the novel view synthesis aspect of our model; there is still room for improvement in view generation quality, especially for dynamic objects, which is currently limited due to the L2 loss. Future work could improve the model's generative capabilities: using diffusion or GAN-like approaches should lead to more plausibly imputed missing information.

We believe that DyST contributes a significant step towards learning neural scene representations from real-world scenes encountered in the wild. This opens the door for exciting down-stream applications, especially when combined with the potential of training on large-scale video collections.

**Author contributions & Acknowledgements.** Maximilian Seitzer: conception, implementation, datasets, experiments, evaluation, writing. Sjoerd van Steenkiste: writing, advising. Thomas Kipf: writing, advising. Klaus Greff: project co-lead, conception, evaluation, writing. Mehdi S. M. Sajjadi: project lead, conception, datasets, early experiments, evaluation, writing.

We thank Aravindh Mahendran for detailed feedback on the manuscript and Alexey Dosovitskiy for fruitful discussions on the project. The authors thank the International Max Planck Research School for Intelligent Systems (IMPRS-IS) for supporting Maximilian Seitzer.

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

# A  APPENDIX

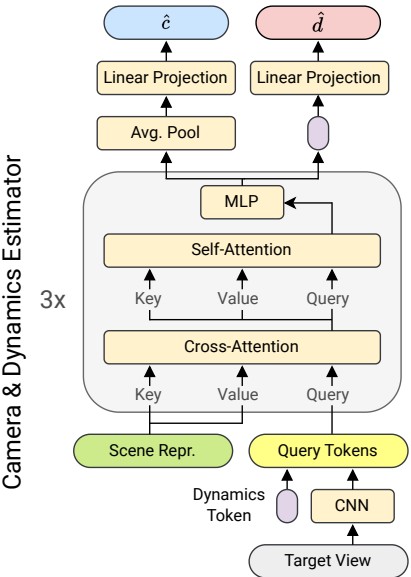

Figure 8: Architecture of joint camera and dynamics estimator. We implement camera and dynamics estimator as a single transformer-based module that iterates cross- and self-attention layers between the target frame and the scene representation. The camera control latent $\hat{c}$ is produced from global average pooling of the output of the last transformer layer, and a low-dimension projection. The dynamics control latent $\hat{d}$ is produced from projecting only the output corresponding to the learned dynamics token.

## A.1  MODEL DETAILS AND HYPERPARAMETERS

Unless mentioned otherwise, the architecture and hyperparameters follow RUST (Sajjadi et al., 2023). We briefly summarize the main details in the following.

**Input.**  The model is trained on images of size $128 \times 128$. We always use 3 input views to compute the scene representation $\mathcal{Z}$.

**Encoder.**  The initial CNN uses 6 layers that reduce the input views to patches of size $8 \times 8$. Flattening, adding a 2D positional encoding, and a linear transformation yields 768 tokens with dimensionality 768. The tokens corresponding to the first input view receive an additional embedding such that they can be differentiated from the remaining views. To compute the scene representation, we then apply a 5 layer transformer encoder with 12 attention heads, MLP with hidden-dimension 1536, and pre-normalization. The scene representation $\mathcal{Z}$ consists of 768 tokens of dimensionality 768.

**Camera & Dynamics Estimator.**  We implement camera estimator and dynamics estimator jointly in a single module that is similar to RUST's pose estimator. See Fig. 8 for an overview of the architecture. Similar to the encoder, a CNN first transforms the target view into patches, but with 8 layers, resulting in a patch size of $16 \times 16$. Adding a 2D positional encoding yields 64 query tokens of dimensionality 768, to which we add a single learned "dynamics token", whose output is used to compute the latent scene dynamics $\hat{d}$. A 3-layer transformer decoder then uses the queries to cross-attend into the tokens of the scene representation corresponding to the first input view $\mathcal{Z}'$, before performing self-attention among the queries themselves. To produce the camera control latent $\hat{c}$, the output of the last transformer layer (excluding the output corresponding to the dynamics token) is global average pooled and linearly projected to $N_c = 8$ dimensions. To produce the dynamics control latent $\hat{d}$, the output corresponding to the dynamics token is linearly projected to $N_d = 8$ dimensions.

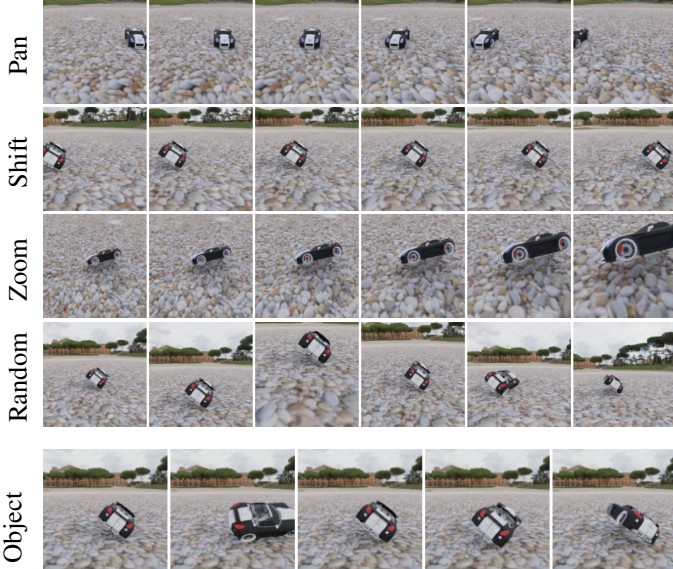

Figure 9: Illustration of motions in the DySO dataset. Row 1–4 show examples of the used camera motions: panning, shifting, zooming and randomly sampled. Row 5 shows an example of how object dynamics: random noise is added to the object's position, and a random rotation up to 90 degrees is applied to the objects pose.

**Decoder.** The queries to the decoder are formed by concatenating the latent camera & dynamics $\hat{c}, \hat{d}$, then adding a 2D positional encoding for each pixel to be decoded. The decoder processes each query independently by cross-attending into the scene representation using a 2-layer transformer (same configuration as in the encoder, but no self-attention). Different from RUST, the queries are linearly transformed to a dimensionality of 768 (instead of 256). Also different to RUST, the outputs of the transformer are mapped to RGB space by a 1-hidden layer MLP with hidden dimensionality 768 (instead of 128).

**Training.** During training, we always use 3 input views and 4 target views per scene, where we render 8192 pixels uniformly across the target views. On the synthetic DySO dataset, we apply latent control swap, where we sample the 4 target views such that capture exactly 2 distinct cameras and dynamics, and the 3 input views are randomly sampled from the remaining cameras and dynamics. We use a batch size of 256, where each data point correspond to a scene. We alternate gradient steps between batches from the DySO and the SSv2 dataset, and train the model end-to-end using the Adam optimizer (with $\beta_1 = 0.9, \beta_2 = 0.999, \epsilon = 10^{-8}$) for 4 M steps. We note that the model has not converged at that point, and we could see PSNR further improving while training longer. The learning rate is decayed from initially $1 \times 10^{-4}$ to $1.6 \times 10^{-5}$, with an initial linear warmup in the first 2500 steps. We also clip gradients exceeding a norm of 0.1. As in RUST, we scale the gradients flowing to the camera & dynamics estimator by a factor of 0.2.

## A.2 DATASETS

**DySO.**

We implement the DySO dataset based on the MultiShapeNet (Sajjadi et al., 2022b) dataset in the Kubric simulator (Greff et al., 2022). The DySO dataset consist of 1M scenes for training, 100K scenes for validation and 10K scenes for testing. Each scene has a randomly selected ShapeNet object (Chang et al., 2015), and consists of a 5 distinct camera positions observing the object moving for 5 steps. This yields 25 views per scene. Images are rendered with ray tracing and on top of 382 complex HDR backgrounds at a resolution of $128 \times 128$.

The cameras are sampled on trajectories with one of 4 motion patterns: horizontal shifts, panning, zooming, and sampling random camera points nearby a fixed point. See Fig. 9 for an illustration. Furthermore, we add small Gaussian noise to all camera positions to make the camera trajectories less predictable. For the object dynamics, for each time step, we randomly jitter the object's position and further apply a random rotation up to 90 degrees to the object. Initially, the object is placed on the floor with a random position and pose.

**SSv2.** The SSv2 dataset showcases short videos of humans performing basic actions with everyday objects from an ego perspective. It consists of roughly 170K training videos, 24K validation videos, and 27K testing videos with 12 frames per second. For training, we sub-sample a clip of up to 64 frames for each video. As the original videos only have a resolution of $100{\times}100$ pixels, we upsample them to $128{\times}128$. All examples shown in the paper are from the test split.

