# OpenReview forum: "DyST: Towards Dynamic Neural Scene Representations on Real-World Videos"
_ICLR.cc/2024/Conference — ICLR 2024 spotlight_

### Official Review · Reviewer_XJou · 2023-10-31

**Soundness:** 3 good
**Presentation:** 3 good
**Contribution:** 3 good
**Rating:** 6
**Confidence:** 4

**Summary:**

This paper proposes Dynamic Scene Transformer that learns latent neural scene representations from monocular dynamic video without any pose information. Different from previous works, this paper mainly focuses on modeling the latent space for dynamic scenes. To achieve this, the authors utilized  a Camera Estimator and a Dynamics Estimator to produce the low-dimensional controllable latents for camera pose and scene dynamics. To separate dynamics from camera pose effectively, the author further design a swap training scheme and establish a multi-view, multi-dynamics dataset synthetic dataset.

**Strengths:**

1. This paper is well-motivated. The primary goal of this paper is the separation of scene dynamics and camera pose, while most of existing works only cover the static scenes.
2. The authors proposes a novel training scheme that disentangles the camera pose from two views under the same camera while containing a moving object, and disentangles the scene dynamic from two views with still objects while under two different cameras. To fulfill this training strategy, the authors also establish a new synthetic data with multi-view, multi-dynamics data.

**Weaknesses:**

1. The method is quite similar to RUST[1]. The encoder, decoder and camera estimator are almost the same as the ones proposed in RUST.
2. Inference procedure. From the method architecture, the target view is required to obtain the camera latents and dynamic latents. In this case,  I wonder if the specific novel view image is needed as the input to generate the novel view?
3. Control the latent code. In Fig7, the authors show the results of controlling camera latent and dynamic latent. The authors could explain how to control the latent code.
4. Some quantitative results on real data should be provided.


[1] RUST: Latent Neural Scene Representations from Unposed Imagery. CVPR, 2023.

**Questions:**

Reproducibility Statement should be put in the appendix.

---

> ### Comment · Reviewer_whxo · 2023-11-18
> **Not seeing rebuttal**
>
> Hi,
>
> The comparison with RUST is a good point and should be more developped by the authors.
> It is indeed mentionned in the paper. Did the authors discovered this work while their method was already implemented ?
> It seems that the motivation to separate between learning the pose and the scene is a difference with RUST work.
>
> Best,

---

> ### Author Response · Authors · 2023-11-20
> **Author Response**
>
> We thank the reviewer for their helpful feedback on our submission and we're happy that they found the approach well-motivated.
> We address the questions below.
>
> **Similarity to RUST**
>
> We fully agree with the reviewer that DyST has a similar network architecture to RUST. This is intentional, as it should be noted that our major contributions are not the network architecture (besides e.g. the Camera & Dynamics estimators), but rather, that the contribution of our work lies in the novel swap-training scheme, co-training between the (newly proposed and to be released) DySO dataset and real-world videos, and the investigation and validation thereof. We believe that focusing on these crucial factors enabling separation into scene content, camera, and scene dynamics is a strength of the approach. While investigations in modifying the network architecture are certainly possible (e.g., to improve render quality even further in future work), they could distract from the novel capabilities of our model that our presented work is focused on, and make it harder to credit the new model capabilities to the novel components that we propose.
>
> **Inference procedure & control of the latent code**
>
> We thank the reviewer for the question. DyST inference can be realized in several ways that do not include a dedicated novel target view. Depending on the use case, one of the input views can serve as the "target" view that is used to get the (initial) control latents. The PCA-analysis demonstrated in Sec. 4.2 and Fig. 6 show how one can control camera and scene dynamics (for any arbitrary video) in intuitive ways. Another possibility is to transfer the camera and / or scene dynamics from one video (e.g., a DySO scene) to another one (e.g., a real video), to have full control over the camera and scene dynamics. Finally, we show an example of scene control over a large number of videos in [1].
>
> [1] https://ml-anon-1.github.io/#cam-ood
>
> **Quantitative results on real data**
>
> We agree with the reviewer that quantitative results on real-world videos would be great, however it is non-trivial, since this would require access to synchronized multi-view videos. We will investigate this possibility and calculate quantitative metrics on real-world videos.
>
>
> **Position of the Reproducibility Statement**
>
> We thank the reviewer for noting this. It is our understanding that the official guide suggests placing the reproducibility statement "at the end of the main text (before references)" [1]. We are happy to correct the placement if there was a misunderstanding on our side.
>
> [1] https://iclr.cc/Conferences/2024/AuthorGuide#:~:text=Reproducibility,been%20made%20to%20ensure%20reproducibility.
>
>
> **Summary**
>
> We hope we have addressed all of the reviewer's questions. Given that our submission was found to be “well-motivated” and rated "good" on all categories (Soundness, Presentation, Contribution), we kindly ask the reviewer to consider updating the score.

---

> ### Author Response · Authors · 2023-11-20
> **Response to Reviewer whxo's comment**
>
> We apologize for the delayed rebuttal – some of the requested experiments needed more time to finish before we could summarize our findings. We address all reviews in the comments above. The point on the similarity with RUST is also addressed above in the comment to reviewer `XJou`. We note that our ablation in Sec. 4.4 (no swap) could be seen as the most similar model to RUST. From that experiment, we can conclude that it achieves a significantly lower PSNR (18.6 vs 26.0), worse LPIPS (0.45 vs. 0.34) and no separation into camera pose and scene dynamics (despite this ablation having a split into camera & scene dynamics components which RUST lacks).
>
> We appreciate the engagement of the reviewer and welcome further comments or questions during the rebuttal phase.

---

### Official Review · Reviewer_Qe3y · 2023-11-01

**Soundness:** 3 good
**Presentation:** 3 good
**Contribution:** 3 good
**Rating:** 6
**Confidence:** 3

**Summary:**

- The authors propose Dynamic Scene Transformer (DyST) which makes a model infer the target view’s control latents (camera pose, scene dynamics) with a pair of corresponding views (different camera pose and scene dynamics from target view).
- Moreover, the authors propose a synthetic dataset, Dynamic Shapenet Objects (DySO), which consists of 5 scene dynamics and 5 camera views for each dynamic scene video to train the DyST.
- By showing qualitative and quantitative results of the experiments on changing control latents, the authors validate that the DyST learns latent decomposition of the space into scene content.

**Strengths:**

- The authors co-train synthetic and real-world datasets to transfer the dynamics and camera control potential of synthetic scenes to natural monocular video and the results shown in Fig. 5 indicate that the model has learned to encode dynamics independently of the camera pose.
- Since there is no architectural difference between camera pose and scene dynamics, the authors propose to enforce separation through a novel latent control swap training scheme, and the results in Fig. 3 demonstrate their method with a high improvement in PSNR scores.

**Weaknesses:**

- [Generalization in different types of motions] Additional experiments are needed to see if the proposed DyST model can generalize to camera poses and scene dynamics that were not seen during training. so, it would be better to provide qualitative results on how the controlled view looks like when horizontal shifts are input after training without horizontal shifts. (DySO’s camera motions consist of 4 horizontal shifts, panning, zooming motions, and random camera points)
- [Cluttered background] Since the backgrounds of the DySO dataset in Fig. 2 and 3 are clean, the authors need to experiment to see if DyST can robustly control the view even when using videos with cluttered backgrounds. In addition, it would be better to have a distance analysis for unclean scenes to see how distinctly it separates camera pose and scene dynamics like the experiment in Figure 5.
- [Quantitative comparison] As the authors mentioned in Sec. 5 Conclusion, unlike NeRF's output, the output of the proposed method has a quality gap, such as objects disappearing or blurring. Therefore, quantitative comparison results such as PSNR and LPIPS between NeRF and DyST are needed.
- [Multiple objects] Also, as mentioned by the authors in the same section, the authors did not provide results for multiple object scenes. It would be helpful to see the results of latent distance analysis in Figure 5, PSNR, and LPIPS in Figure3 for multiple object scenes.

**Questions:**

- The latent control swap training scheme needs 3 input views. It would be helpful why 3 input views are needed and how the performance changes with less or more than 3 input views.
- It would be better if the authors discuss why it needs the contrastiveness metric and what the authors are trying to show with the swap in Table 1.

---

> ### Author Response · Authors · 2023-11-20
> **Author Response**
>
> We thank the reviewer for their valuable feedback on our submission, and we appreciate that the reviewer acknowledges that DyST learns to encode dynamics independently of the camera pose and that our contributions lead to high improvements in PSNR scores. We address the questions below.
>
>
> **Generalization to novel types of motions**
>
> In theory, DyST will only learn to model camera motion and scene dynamics that exist in the data, as this is strongly encouraged by the model architecture. However, in practice, two types of out of distribution (OOD) generalization are observed:
> Between-scene OOD generalization. The model learns dynamics that may exist only in a subset of videos, and at inference time, these dynamics can then be applied to other videos too.
> Full-dataset OOD generalization. Since the model learns a decomposed representation of camera and scene dynamics, it exhibits OOD generalization in terms of novel dynamics that did not exist in the dataset. For example, the rotation of the camera demonstrated in [1] is neither existent in DySO, nor did we find examples in SSv2 that show this kind of camera motion.
> For the camera-ready version, we can try to include a further experiment on a DySO variant that e.g. does not exhibit any horizontal shifts to see if the model generalizes to capture this at test time. That said, we believe that the result in [2], showing the rotating motion which was non-existent in DySO nor SSv2, already shows a much significantly stronger generalization capability in the same vein.
>
> [1] https://ml-anon-1.github.io/#cam-ood
>
>
> **DyST on cluttered backgrounds & distance analysis**
>
> We thank the reviewer for bringing this to our attention. Indeed the background of some scenes from DySO in our figures may look simple, however, DySO includes a large number of nontrivial HD backgrounds that include clutter. We have added many more results in [1] to show a better representation of the dataset. Further, the SSv2 dataset contains a large number of videos with cluttered backgrounds, where the model still exhibits robust view control. To highlight this further, the video in [2] includes videos from SSv2, and more scenes that we recorded ourselves, with cluttered backgrounds, e.g. additional static objects in the background.
>
> We note that the distance analysis on the left side of Fig. 5 was calculated over a large number DySO scenes, including ones with cluttered backgrounds. The video on the right that we recorded ourselves already includes quite a lot of clutter, please note the static objects on the table. We also have included more distance analysis results in [3] for convenience.
>
> [1] https://ml-anon-1.github.io/#dyso
>
> [2] https://ml-anon-1.github.io/#cam-ood
>
> [3] https://ml-anon-1.github.io/#latent-distances
>
>
> **Quantitative comparison**
>
> We thank the reviewer for suggesting a comparison to NeRF. We'd like to emphasize that common high-quality NeRF videos have been generated in quite different setups, with access to high-quality camera poses and usually for static scenes. Furthermore, NeRF methods require expensive optimization per scene (or in this case, per video), and do not generalize in real-time to novel videos. Finally, NeRF methods usually overfit a network to the specific scene and therefore neither provide a handy latent representation of the scene content (in our case, the SLSR), nor scene dynamics.
>
> With the caveats above, we agree with the reviewer that a comparison would still be valuable. We identified the strongest NeRF model that works on dynamic videos without requiring camera poses, Robust DynRF, and use the official implementation from [1]. We train the Robust DynRF for 50k steps on 6 scenes from SSv2, and find that it achieves an average PSNR of 26.1 and LPIPS of 0.34 on those scenes. In comparison, DyST achieves a significantly better PSNR of 27.9 and LPIPS of 0.18. Notably, we find that Robust DynRF produces artifacts around moving objects (see [2]), and is not able to generalize out of distribution (OOD) when moving the camera out of the frame of the video, whereas DyST is able to smoothly fill-in a reasonable background. Moreover, training the model on a single scene takes around 3.5 hours on a 4080 GPU, whereas a trained DyST model provides inference in a few milliseconds. We conclude that in the complex setting we target in this work, DyST has significantly better render performance than SotA NeRFs, while our model provides superior OOD performance, vastly faster inference, and handy latent representations of the videos.
>
> [1] https://github.com/facebookresearch/robust-dynrf/
>
> [2] https://ml-anon-1.github.io/#nerf-comparison

---

> ### Author Response · Authors · 2023-11-20
> **Author response (2nd part)**
>
> **Multiple objects**
>
> We thank the reviewer for bringing this up. Our statement in Sec. 5 (conclusion) was meant to address multiple independent moving objects in a single video, we have improved the formulation in the draft. Indeed, closer inspection of the results we have provided reveals that a number of our real-world videos already show multiple moving objects, even in Fig. 5 on the right, with the hand and the cup being two moving objects. As the figure shows, the model can, to some extent, therefore generalize to multiple objects. Running an experiment on synthetic DySO scenes with multiple objects involves creating a new dataset and will therefore take more time, however we will try to include this in the final version of the manuscript.
>
>
> **Number of input views for control swap**
>
> We thank the reviewer for proposing this experiment.
> It should be noted that the proposed control swap scheme has no requirements on the number of input views. In early experiments, we ran experiments with fewer (down to 1) or more input views, and found that the model largely behaved the same, with the expected difference that reconstruction quality would degrade or improve subtly depending on how much information the model receives. We chose 3 input views at training time to strike a good balance between efficiency and quality. It should be noted that the number of input views to DyST can be varied at inference time (in particular, it can be different from training), however the comparison would not be apples-to-apples, since more information would be given to the model.
> For completeness, we note that there is a requirement on the number of target views during training: a minimum of 2, so a swap can be performed. However, this only applies on the synthetic dataset for co-training. On the real-world dataset (SSv2 in our case), even 1 input view and 1 target view would suffice for training.
>
>
> **Contrastiveness metric & Swap in Tab. 1**
>
> With the contrastiveness metric and the different swap training scheme ablations, we aim to investigate quantitatively how cleanly the model separates camera motion from scene dynamics, i.e. the first question in Sec. 4.2: "First, does our training with latent
> control swapping induce a meaningful separation of camera pose and scene dynamics on synthetic data?"
> The contrastiveness measures the similarity between pairs of camera and scene dynamics latents for scene views that exhibit various combinations thereof. Intuitively, R^cam will be exactly zero if the latent camera pose is perfectly consistent for views rendered for the same camera, fully independently of the scene dynamics (and vice versa). In practice, the camera and scene dynamics latents are never perfectly separated and independent of each other, therefore the R^cam and R^dyn values are positive.
>
> Tab. 1 analyzes the results of different training schemes for the proposed DyST model. We show that our proposed swap method leads to the cleanest separation between camera and scene dynamics, and that the same model trained without our swap scheme ("No swap") learns practically no separation, thereby providing quantitative evidence that our training scheme is helpful.

---

### Official Review · Reviewer_whxo · 2023-11-01

**Soundness:** 3 good
**Presentation:** 3 good
**Contribution:** 3 good
**Rating:** 8
**Confidence:** 3

**Summary:**

The goal of this paper is to estimate motion and object pose and shape from monocular videos using a latent neural representation.
Two modules are trained for camera parameter and object position and shape estimations. Then, these modules are then used as input to a third decoder module to generate novel views. To ensure the specialization of the modules, the authors proposed a training procedure where the data is organized to enforce which information is learned by each module. Evaluations are conducted on a newly created dataset which was used to train the model and qualitative results are presented with motion extraction, novel view generation, video manipulations where novel camera motions or object movements are generated.

**Strengths:**

Compared to the state of the art, this work investigates the more difficult setting of estimating moving objects and moving cameras only from a few motions pictures and a monocular camera. Moreover, they remove the assumptions of training one model for each scene.

The separation of 3D structure estimation and camera motion is an interesting property of the model. The training tricks illustrated by Eq. 5 and Eq. 6 provide an practical way of enforcing this while still retaining the benefit of end to end training.

Although the videos are simple and are still far from the complexity of most real world data,it is still a good compromise as a next step toward more mature systems. Experiments shown in Fig.3 to assess the specialization of the different modules are convincing, it is also supported by the qualitative results shown in the videos in the supplementary material on video manipulation and image synthesis.

Experiments seems reproducible, given code and training parameters, and available datasets will be provided.

**Weaknesses:**

The amplitude of the motion would probably limits the accuracy of the method. In Fig.7 the motion is tiny, and this is not evaluated by the authors. Although the encoder and decoder architectures are rather small for the "simple" cases covered by the paper, I have concerns on the scalability of this method to more real cases and more complex motions.

**Questions:**

Is the latent dynamic space somehow interpretable, and is it possible to generate one instance based on the object position in space, or does the latent space always be inferred from an existing image contained in the processed sequence ?
In the later case, this means the approach cannot be used to recreate dynamics that do not exist yet in the data ?

---

> ### Author Response · Authors · 2023-11-20
> **Author response**
>
> We thank the reviewer for their very positive feedback on our submission. We appreciate that the reviewer agrees the unsupervised setting is quite difficult, that the proposed method presents a good step toward more scalable models, and that the results are convincing and reproducible. We address the questions below.
>
> **Limited motion in Fig. 7 & scalability**
>
> We thank the reviewer for pointing this out. In [1], we include further qualitative results on videos that contain more complex scenes and motion to show that the model generalizes to larger motion, even for videos that we recorded ourselves (i.e., they are not from SSv2). We observe that the model generalizes to videos outside of SSv2 and that it scales gracefully to more significant camera motion and complex scenes. On the topic of further scalability, we believe that future work will greatly benefit from the inclusion of a stochastic component in the model, e.g. a diffusion decoder instead of the L2 loss, to lead to sharper reconstructions.
>
> [1] https://ml-anon-1.github.io/#cam-ood
>
>
> **Interpretability of the latent space & controllability**
>
> Thank you for the question. A PCA decomposition of the aggregate posterior scene dynamics space reveals principal components that have an intuitive interpretation, e.g. object rotations, translations, and height (see Fig. 6 in the paper). Note that these principal components have been calculated over a large number of scenes and they can therefore be used to control the scene dynamics of any scene. To initialize the latent scene dynamics, one can use one of the input views, so no dedicated target view is required to this end.
>
> In theory, DyST will only learn to model camera motion and scene dynamics that exist in the data, as this is strongly encouraged by the model architecture. However, in practice, two types of out of distribution (OOD) generalization are observed:
> Between-scene OOD generalization. The model learns dynamics that may exist only in a subset of videos, and at inference time, these dynamics can then be applied to other videos too.
> Full-dataset OOD generalization. Since the model learns a decomposed representation of camera and scene dynamics, it exhibits OOD generalization in terms of novel dynamics that did not exist in the dataset. For example, the rotation of the camera demonstrated in [1] is neither existent in DySO, nor did we find examples in SSv2 that have noticable camera motion of this kind.
>
> [1] https://ml-anon-1.github.io/#cam-ood

---

### Author Response · Authors · 2023-11-20
**General comment**

We thank all reviewers for their positive feedback and valuable suggestions. We appreciate that the reviewers find our research “well-motivated”, that we investigate a “difficult setting”, and that our experiments showing camera-dynamics separation are “convincing” and yield a “high improvement in PSNR scores”. We also agree with reviewer `whxo` that our model is a “step towards more mature systems”. Moreover, we are happy that all reviewers rate all three categories as "good" (Soundness, Presentation, Contribution).

We present the new results from this rebuttal on the following anonymous website, https://ml-anon-1.github.io/ in order to facilitate inspection of the videos. The paper will be updated appropriately.

A brief summary of our new experiments in this rebuttal:

* **Synchronized control & generalization to novel motions and self-recorded videos** (see https://ml-anon-1.github.io/#cam-ood): we find that DyST handles larger motions in the video well and gracefully generalizes to OOD camera motions.
* **Extended latent distance analysis from Fig. 5 to more real world videos** (see https://ml-anon-1.github.io/#latent-distances): we find more evidence that DyST learns a clean separation of the latent space into camera and scene dynamics
* **Quantitative and qualitative comparisons to Robust DynRF**, a state-of-the-art NeRF method for unposed monocular dynamic video (https://ml-anon-1.github.io/#nerf-comparison): we find that DyST performs overall better in render quality, with Robust DynRF often producing artifacts on camera & object motions. It should be noted that Robust DynRF is significantly more compute intensive and most importantly, contrary to DyST, it does not provide useful representations for the videos.

We address further questions by the reviewers directly below.

---

### Meta-Review · Area_Chair_i5NP · 2023-12-11

**Metareview:**

The paper studies the problem of dynamic neural scene representation learning from monocular videos. The key idea is to learn a latent decomposition of scene content and per-view scene dynamics. All reviewers are positive about the approach and appreciate the results on challenging scenarios. In the rebuttal, the authors provide more results and address the concerns raised by reviewers. The area chair, therefore, recommends to accept the submission.

**Justification For Why Not Higher Score:**

The technical novelty is moderate and there are still concerns on the ability to address complex real scenes, in terms of scene geometry complexity and video capturing trajectory.

**Justification For Why Not Lower Score:**

The reviews are all positive and the results show a clear progress on the topic. Why lower?

---

### Decision · Program_Chairs · 2024-01-16

Accept (spotlight)